# The origin of exceptionally large ductility in molybdenum alloys dispersed with irregular-shaped La$_2$O$_3$ nano-particles

Yujie Chen[1,2,8], Yan Fang[1,3,8], Pengming Cheng[4,8], Xiaoxing Ke [5], Manchen Zhang[5], Jiawei Zou[1], Jun Ding [6] ✉, Bozhao Zhang[6], Lin Gu [2] ✉, Qinghua Zhang [7], Gang Liu [4] ✉ & Qian Yu [1] ✉

Molybdenum and its alloys are known for their superior strength among body-centered cubic materials. However, their widespread application is hindered by a significant decrease in ductility at lower temperatures. In this study, we demonstrate the achievement of exceptional ductility in a Mo alloy containing rare-earth La$_2$O$_3$ nanoparticles through rotary-swaging, a rarity in Mo-based materials. Our analysis reveals that the large ductility originates from substantial variations in the electronic density of states, a characteristic intrinsic to rare-earth elements. This characteristic can accelerate the generation of oxygen vacancies, facilitating the amorphization of the oxide-matrix interface. This process promotes vacancy absorption and modification of dislocation configurations. Furthermore, by inducing irregular shapes in the La$_2$O$_3$ nano-particles through rotary-swaging, incoming dislocations interact with them, creating multiple dislocation sources near the interface. These dislocation sources act as potent initiators at even reduced temperatures, fostering diverse dislocation types and intricate networks, ultimately enhancing dislocation plasticity.

Whether a material is ductile or brittle is determined by its mechanism for releasing strain energy, either through dislocation motion and interaction or crack nucleation and propagation. Body-centered cubic (BCC) metallic materials, known for their high strength, often exhibit limited ductility due to insufficient dislocation interactions and multiplication[1-3]. Commercial-purity molybdenum (Mo) metal, for example, typically displays only a small percentage of tensile elongation at room temperature[4]. Ductility decreases further as temperature

drops, severely limiting its engineering applications[5,6]. Clearly, when dislocation plasticity is facilitated and/or crack nucleation is impeded, the materials deform through dislocation activities, leading to the increase of ductility[1].

Rare-earth elements, as vital strategic resources, exert a transformative influence on material properties[7-11]. Although their presence may not significantly contribute to weight, value, or volume, it is often indispensable for optimal material performance. In less-ductile BCC

[1]Center of Electron Microscopy and State Key Laboratory of Silicon and Advanced Semiconductor Materials, Department of Materials Science and Engineering, Zhejiang University, Hangzhou 310027, China. [2]Beijing National Center for Electron Microscopy and Laboratory of Advanced Materials, School of Materials Science and Engineering, Tsinghua University, Beijing 100084, China. [3]Department of Mechanical Engineering, The University of Hong Kong, Hong Kong 999077, China. [4]State Key Laboratory for Mechanical Behavior of Materials, School of Materials Science and Engineering, Xi'an Jiaotong University, Xi'an 710049, China. [5]Beijing Key Laboratory of Microstructure and Properties of Solids, College of Materials Science and Engineering, Beijing University of Technology, Beijing 100124, China. [6]Center for Alloy Innovation and Design, State Key Laboratory for Mechanical Behavior of Materials, Xi'an Jiaotong University, Xi'an 710049, China. [7]Beijing National Laboratory for Condensed Matter Physics, Collaborative Innovation Center of Quantum Matter, Institute of Physics, Chinese Academy of Sciences, Beijing 100190, China. [8]These authors contributed equally: Yujie Chen, Yan Fang, Pengming Cheng. ✉e-mail: dingsn@xjtu.edu.cn; lingu@mail.tsinghua.edu.cn; lgsammer@mail.xjtu.edu.cn; yu_qian@zju.edu.cn

structured metals and alloys, adding rare-earth oxides like $CeO_2$[12], $La_2O_3$[13-15], and $Y_2O_3$[16-18] showed promise in simultaneously enhancing strength and ductility. Prior researchers hypothesize that rare-earth oxides not only promote grain nucleation, but the oxygen vacancies at the surface adsorb detrimental impurities such as O, N atoms et. al., thereby reducing grain boundary-induced crack nucleation[15,19]. However, this hypothesis lacks empirical evidence and fails to account for the increased intergranular dislocation activity. Recently, we realized that rare-earth elements, owing to the strong correlation between electrons in f-orbitals and their multivalence nature, exhibit a range of exotic electronic states[20]. In the field of chemistry, during the complex redox reactions[21] or thermal processes involved in rare-earth oxide synthesis[22], oxygen partial pressure undergoes specific changes near the growth interface where diffusion is facilitated, allowing for highly tunable oxygen vacancy concentrations. The combined impact of these alterations in electronic and geometrical structures may lead to crystal structure collapse and ordered-to-disordered structural transitions[23,24]. This insight prompts us to consider whether the introduction of rare-earth elements in alloys results in distinct precipitate-matrix interfaces that influence dislocation behavior in distinctive ways.

In this work, through integrated differential phase contrast-scanning transmission electron microscopy (iDPC-STEM) and electron energy-loss spectroscopy-STEM (EELS-STEM) analysis, we first ascertained that precise control of the interfacial reaction during even conventional liquid-solid mixing effectively facilitates the formation of an ultrathin amorphous interfacial layer between $La_2O_3$ particles and the Mo matrix, which exhibits a propensity for vacancy absorption and modification of dislocation configurations. This phenomenon is truly closely linked to the alterations in the electronic density of states within La. To further harness this potent effect, we adopted rotary-swaging and produced a Mo alloy containing irregular-shaped $La_2O_3$ (0.6 wt.%) nanoparticles with the amorphous interface. Combining three-dimensional tomography and in-situ TEM mechanical testing, we identified that this distinctive interface plays a crucial role in enhancing dislocation activities. The irregular-shaped amorphous interface is capable of promoting the formation of numerous and diverse dislocation sources at the particle-matrix interface, significantly bolstering dislocation plasticity. Consequently, the resulting alloy demonstrates a strength of approximately 783 MPa and an elongation of about 37.5 % at room temperature. At −50 °C, it even demonstrates an elongation at fracture of approximately 37% and a yield strength of approximately 1014 MPa, surpassing the performance of other Mo alloys.

## Results

The sintered Mo-$La_2O_3$ (0.6 wt.%) alloy was produced using power metallurgy, with Mo-$La_2O_3$ alloy powders obtained through a liquid-solid mixing method. Subsequently, the powders were squeezed into a cylindrical compact (90 mm in diameter) and then sintered at 1850 °C for 4 h in a dry hydrogen atmosphere. Figure 1a displays a typical high-angle annular dark-field STEM (HAADF-STEM) image of the as-prepared alloy. It is observed that the nearly spherical $La_2O_3$ nanoparticles are mostly dispersed within grains. To reveal the atomistic interfacial structure, we employed HAADF-STEM along with the advanced iDPC-STEM technique, which enables sub-angstrom resolution imaging of both light and heavy atoms. Figure 1b and S1 present the atomic-resolution HAADF-STEM and iDPC-STEM images viewed along the [101] zone axis of $La_2O_3$. Of particular significance is the discovery of an ultrathin amorphous layer at the particle-matrix interface, with a measured thickness of approximately 1.5 nanometers. This is further confirmed by the fast Fourier Transform image in the upper right corner of Fig. 1b, where a typical amorphous ring characteristic indicative of its amorphous nature is clearly discerned.

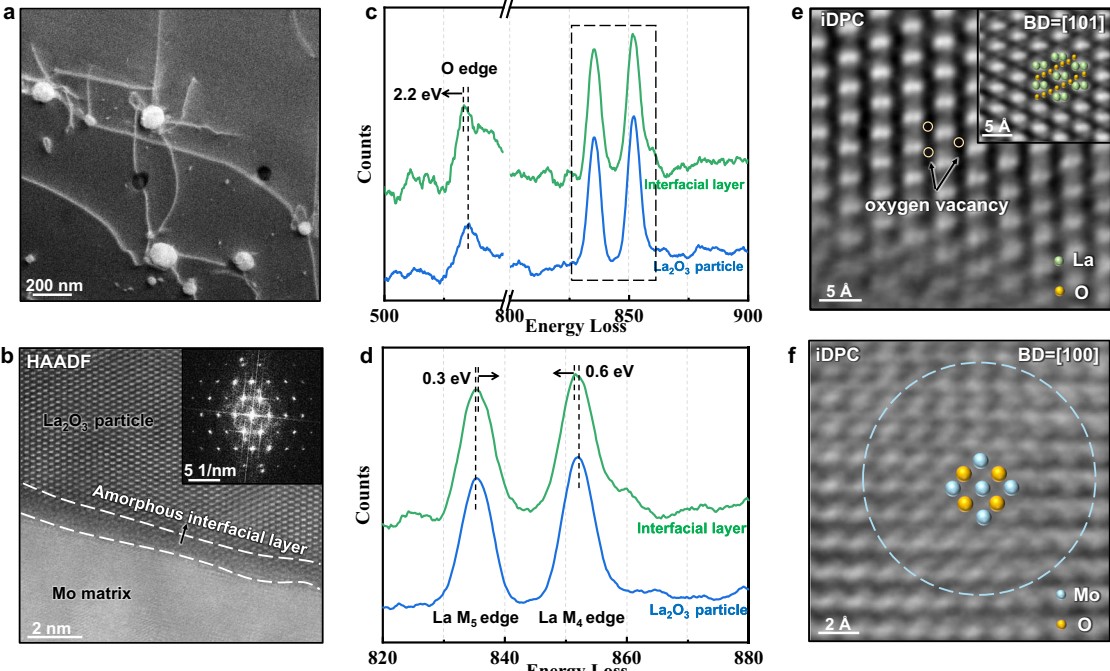

**Fig. 1 | Multi-scale structure of the sintered Mo-$La_2O_3$ alloy. a** HAADF-STEM image of the typical $La_2O_3$ nanoparticles and the dislocations at the interface. **b** HAADF-STEM image showing the atomic structure of the $La_2O_3$ nanoparticle in a [101] zone axis, an ultrathin amorphous interfacial layer is clearly seen. The fast Fourier transforms image in the upper right corner also displayed an amorphous ring accordingly. **c** Oxygen K edges and La M edges were recorded from different areas in the $La_2O_3$ particles. The corresponding image for this EELS mapping is shown in Fig. S2. Blue is from the internal $La_2O_3$ particle. Green is from the area near the interface. **d** Fine structure of the La M edges, obtained and enlarged from the spectrum bounded by the dashed rectangle in **c**. **e** The iDPC-STEM image displaying the oxygen vacancy in $La_2O_3$, green spheres represent La atoms and yellow spheres represent O atoms, while open yellow circles indicate typical oxygen vacancies. **f** The oxygen clusters in the near interface region in the Mo matrix. Yellow spheres represent O atoms, while light blue spheres represent Mo atoms.

To investigate the potential alterations in electronic structures, we then employed the EELS-STEM technique, which possesses the ability to obtain the density of electronic state and valence state information at atomic resolution. Figure 1c shows the O K edges and La M edges recorded from two regions. Region I is in the internal $La_2O_3$ particle (as marked by a blue rectangle in Fig. S2), and Region II (green rectangle) is in proximity to the amorphous layer. We observe substantial shifts in both the O K edge and La M edge from Region I to Region II. Specifically, the O K edge appears a 2.2 eV shift to low energy from Region I to Region II, indicating the existence and considerable concentration of oxygen vacancies[25,26]. To maintain the electron balance, the electron structure of La changes accordingly. In the magnified images of the edge energy-loss near-edge structures (ELNES) of La $M_{5,4}$ edge ($3d{\rightarrow}4f$) excitation edge in Fig. 1d, La $M_5$ edge (the former) appears a 0.3 eV shift to high energy from Region I to Region II, while La $M_4$ edge (the latter) appears a 0.5 eV shift to low energy, which reflects the changes of the $4f$ unoccupied density of state of La atom[27,28].

Meanwhile, oxygen vacancies are clearly observed in close proximity to the amorphous interfacial layer within the nanoparticles. The iDPC-STEM image of the perfect lattice of $La_2O_3$ is shown in the enlarged image at the upper right corner of Fig. 1e for comparison. It becomes apparent that oxygen atoms are locally absent in the oxides near the amorphous interface as highlighted by the pale-yellow open circles in the iDPC-STEM image (acquired from the area demarcated by the orange square, region III in Fig. S1). Additionally, as shown in the magnified iDPC-STEM image in Fig. 1f of Region IV in Fig. S3 (all viewed along the zoon axis [100] of the Mo matrix), oxygen atoms tend to aggregate into nano-sized clusters near the particle-matrix interface in the Mo matrix. The oxygen atoms, indicated by yellow spheres, occupy octahedral interstitial sites within the Mo lattice, marked by light blue spheres.

The above results demonstrate that the significant change in the electronic density of states truly exists from the $La_2O_3$ oxide interior to the interface, which is primarily responsible for the interface amorphization. Other oxides, for instance $Al_2O_3$ shown in Fig. S4, usually display crystalized incoherent interfaces, consistent with previous report[29]. It is also noted that the amorphization is quite sensitive to the control of interfacial reactions. Long-time annealing or hot isostatic pressing sintering may eventually lead to the crystallization of this ultrathin layer. Our in-situ TEM mechanical testing further elucidates the pivotal role of such an amorphous interface in enhancing dislocation interactions and multiplication. As shown in Fig. 2a and Supplementary Movie 1, when an edge dislocation (with a Burgers vector [111]) encounters two nearby particles, instead of looping around[30,31], a portion of the dislocation is absorbed, leaving three dislocation segments. Dislocation segments I and III (marked by blue dashed lines) are anchored at one end of the interface, while segment II (marked by yellow dashed lines) is pinned at both particles. Dislocation segment II continues to advance, giving rise to two screw dislocations firmly anchored at the interface. These screw dislocations exhibit limited mobility, significantly impeding the movement of the connected edge dislocation (segment II). Consequently, this interaction triggers the cooperation of different dislocation components, promoting various dislocation interactions and the operation of single-arm dislocation sources, as illustrated in Fig. S5.

This dislocation-particle interaction is then computationally verified via Molecular Dynamics (MD) simulations at room temperature. The detailed information of the simulation is described in "Methods". A BCC Mo configuration with a 20 nm amorphous particle (simulated using amorphous MoPd alloy for modeling the effect of amorphous interface) was employed (Fig. 2b). Under applied loading, an edge dislocation was inserted and glided on the $(1\bar{1}1)$ plane. Snapshots in Fig. 2b illustrate the configuration during increasing shear strain (Supplementary Movie 2). When the dislocation meets the amorphous

particle, the interacting segment dissolves into the amorphous interface, a known high-capacity dislocation sink[32,33]. Despite this interaction, the dislocation continues to glide, with one end remaining anchored to the amorphous interface. This simulation corroborates experimental observations, confirming the distinct behavior of dislocations near amorphous interfaces.

Furthermore, it is noteworthy that the $La_2O_3$ nanoparticles exhibit no specific orientation relationships with the Mo matrix. The absence of a preferred orientation allows the particles to assume various shapes without inducing interfacial cracks. One might question whether, in the case of a corrugated interface, dislocation segments would become "float" and immobilized at the interface's ridges instead of being absorbed. This hypothesis was initially proved by our computational simulation. We replaced another spherical particles containing two notches at the interface, as shown in Fig. 2c and Supplementary Movie 3. As the gliding dislocation interacts with the particle, it leaves behind two dislocation segments at the notch. If more irregular shapes (e.g., holes) exist on the interface, more dislocation segments would be created after the gliding of dislocations. These dislocation segments, with both ends pinned at the edge of the notch, function as typical Frank-Read dislocation sources upon loading. In this way, one dislocation can be divided into many segments during the dislocation-particle interactions.

To harness this mechanism, we employed the process of rotary-swaging to treat the sintered $Mo\text{-}La_2O_3$ alloy. The sintered $Mo\text{-}La_2O_3$ was rotary-swaged into a rod at successive decreasing temperatures from 1000 to 600 °C and annealed at different temperatures, which subjects the material to compression in all directions, inducing substantial deformation in both the particle and the Mo matrix (detailed in Methods). Fig. S6 provides a picture of $Mo\text{-}La_2O_3$ alloy bar after the process of rotary-swaging. Given the relative forces acting on specific crystallographic planes may differ, large particles tend to fracture into smaller ones and the nanoparticles within the grain are deformed into irregular shapes. As shown in the HAADF-STEM image in Fig. 2d, the $La_2O_3$ nanoparticles exhibit ellipsoid shapes in the projected view, rather than being spherical. Concurrently, there is slight variability in the thickness of the amorphous layer (see Fig. S7). Furthermore, as shown in Fig. 2d, dislocations curled up and tangled around the particles severely in the initial microstructure of this alloy processed by rotary-swaging.

Since the TEM images only provide two-dimensional projections of the actual structure, we conducted 3D tomography analysis[34] to characterize the morphology of these $La_2O_3$ nanoparticles. This involved capturing a series of HAADF images, with each image taken at one-degree intervals while tilting the x-axis of the sample holder under a constant two-beam condition[35–37]. All the images were processed using Tomviz software to generate a 3D tomographic reconstruction of the particles. As depicted in Fig. 2e, Fig. S8, and Supplementary Movies 4 and 5, the results of the 3D tomography unequivocally clarify that the particles exhibit irregular shapes irrespective of their sizes. More precisely, these particles are neither spherical nor faceted, instead, they feature conspicuous ridges and valleys that extend widely around the particle-matrix interface.

The produced $Mo\text{-}La_2O_3$ alloy demonstrates superior tensile properties in comparison to other Mo metals and alloys. We first compared the tensile properties of pure Mo and the $Mo\text{-}La_2O_3$ alloy processed via rotary-swaging. The tensile samples are dog-bone shaped with gauge sizes of 25 mm in length and 5 mm in diameter. Figures 3a, b show the tensile engineering and true stress-strain curves of the tested alloys at different temperatures, respectively. At ambient temperature (approximately 20 °C), the tensile yield strength ($\sigma_y$) of the $Mo\text{-}La_2O_3$ alloy processed by rotary-swaging measures approximately 783 MPa, with a corresponding elongation at fracture of approximately 37.5%. These values are notably superior to those of commercial-purity Mo metal[38]. Obvious work hardening is also

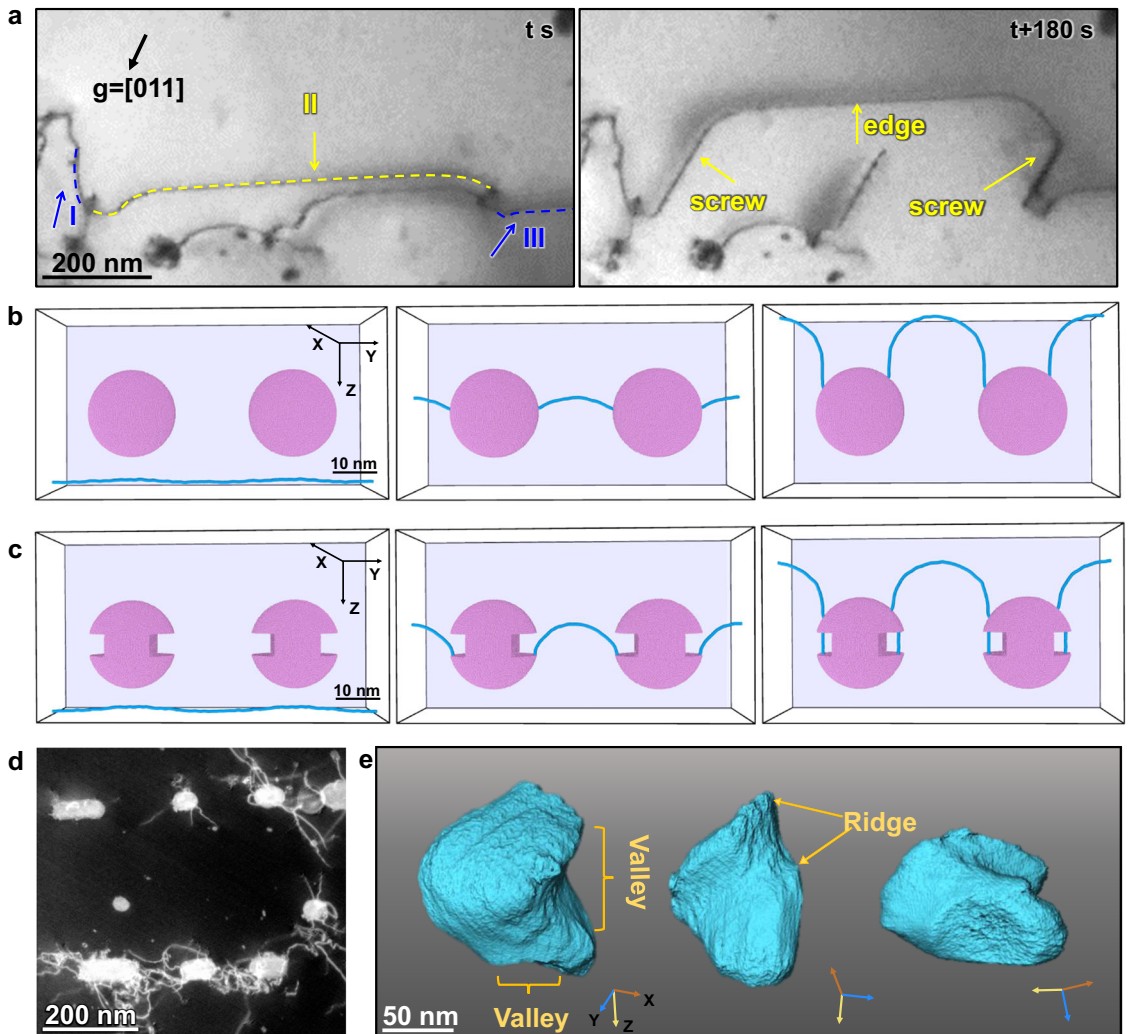

**Fig. 2 | The interaction of dislocations and amorphous interface and the formation of irregularly shaped La₂O₃ particles after rotary-swaging treatment. a** In-situ TEM tensile testing on sintered Mo-La₂O₃ alloy at room temperature. The two particles split the dislocation into three segments, denoted as I (blue arrow), II (yellow arrow), and III (blue arrow), and cause dislocation segment II to form a screw dislocation component. **b, c** Molecular dynamics (MD) simulation showing edge dislocations in Mo gliding across the embedded amorphous MoPd particles with (**b**) the perfectly spherical shape and (**c**) two additional notches. The axes X, Y, Z correspond to the directions of [1$\bar{1}$1], [111], and [11$\bar{2}$], respectively. **d** The typical configuration of particles after rotary-swaging. The morphology of the particles becomes irregular and the entanglement of the dislocations becomes more significant. **e** The three-dimensional tomography of the La₂O₃ particles. The irregular-shaped particles have considerable ridges and valleys at the interface, as marked by yellow arrows.

obtained at both room and low temperatures. As demonstrated in Fig. S9, the Mo-La₂O₃ alloy treated with rotary-swaging also has better strength and elongation than that treated by rolling[14]. We further directly compared the properties of the Mo-La₂O₃ alloy processed by rotary-swaging to other previously reported high-performance Mo alloys containing different oxides and grain sizes. As illustrated in Fig. 3c, the strength-ductility trade-off is obvious in those alloys at even room temperature, while our Mo-La₂O₃ alloy processed by rotary-swaging displays high strength and significant tensile elongation, distinguishing it from the aforementioned alloys. This excellent combination of strength-ductility remains even at −50 °C (Fig. 3a). The material's yield strength reaches 1014 MPa, and the elongation at fracture reaches approximately 37% with obvious work hardening. This low-temperature ductility surpasses that of previously reported Mo and Mo alloys (Fig. 3d).

The large elongation achieved here was found to be highly related to the enhanced dislocation activities. We employed the dislocation tomography technique to reconstruct the actual dislocation structure around the La₂O₃ nanoparticles in this Mo-La₂O₃ alloy after fracture.

The series of HAADF-STEM images were captured by using a double-tilt tomography holder at a tilting range of −56° to 45° with an increment of 1°. We maintained a two-beam condition with **g** = [002] to ensure the visibility of dislocations during sample tilting. Supplementary Movie 6 presents a representative 3D dislocation tomographic reconstruction, while Fig. 4a displays three images extracted from the reconstruction at different tilting angles. It reveals the formation of a complex dislocation structure around the La₂O₃ nanoparticles. Numerous dislocations are observed with one end pinned to the interface. Dislocation segments with both ends pinned at the interface are also found, primarily at the interfacial ridges (marked by orange arrows). Furthermore, substantial dislocation entanglements are formed on or around the particles (as marked by blue arrows).

To enhance the clarity of the visual representation, we employed Chimera software to reconstruct the three-dimensional dislocation structure based on the experimental images. By plugging in the 3D tomography of the particle and dislocations, we attained a simulated 3D dislocation structure, as displayed in Fig. 4b. The intricate interactions between dislocations and particles initiate interactions among

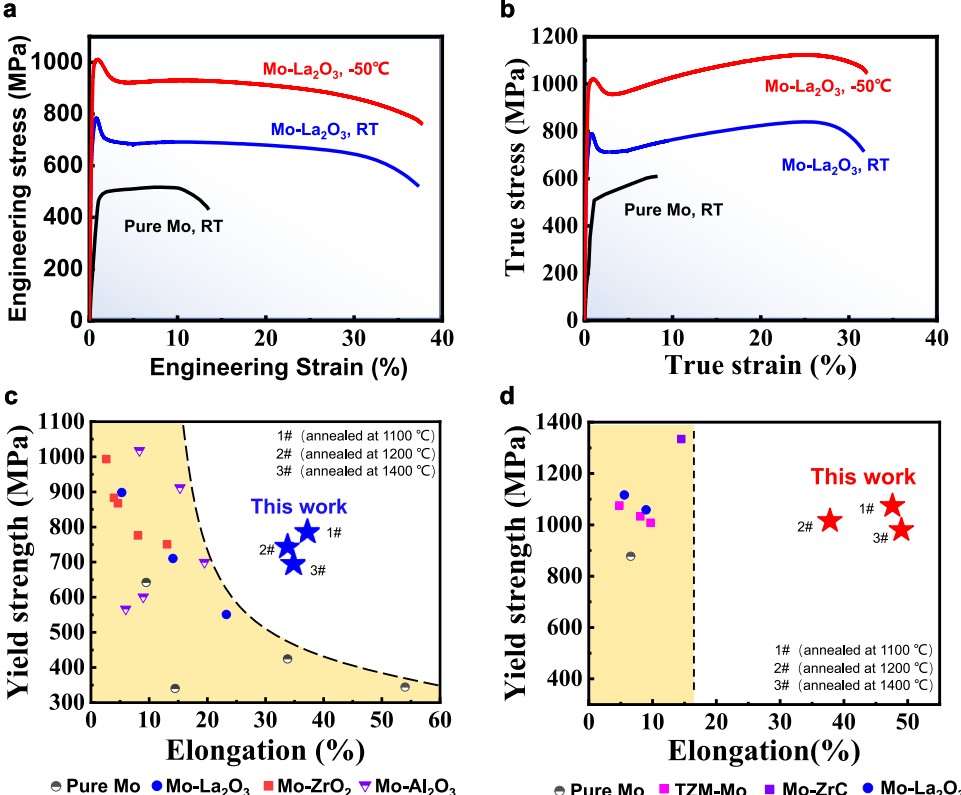

**Fig. 3 | The tensile properties of our Mo-La2O3 alloy after rotary-swaging treatment at different temperatures. a, b** The engineering and true strain-stress curves of pure Mo[38] at room temperature (black line), Mo-La₂O₃ alloy (blue line) at room temperature, and −50 °C (red line), respectively. RT: room temperature.

**c, d** Yield strength versus total tensile elongation of Mo-based alloys at room temperature (Pure Mo[4,44–46], Mo-La₂O₃[4,14,15], Mo-ZrO₂[47], Mo-Al₂O₃[29]) and −50 °C (Pure Mo[4], TZM-Mo[4,48], Mo-ZrC[45], Mo-La₂O₃[4,48]), respectively. 1, 2, and 3# represent the Mo-La₂O₃ alloys annealed at 1100, 1200, 1400 °C, respectively.

dislocations in different slip directions, and planes are enhanced. In Fig. 4c and Fig. S10, the index of dislocation Burgers vectors (**b**) obtained by **g·b** analysis (**g**, vector diffraction) provides further insight into the dislocations pinned at specific particles. These dislocations exhibit a range of **b** values, including 1/2[111], 1/2[1$\bar{1}$1], 1/2[1$\bar{1}\bar{1}$], and [001]. Additionally, the dislocations predominantly consist of mixed dislocations containing both edge and screw components.

The TEM characterization mentioned above has confirmed the advantageous role of irregularly shaped La₂O₃ nanoparticles with amorphous layers in promoting dislocation interactions and dislocation multiplication. To reveal the dynamic of the complex dislocation-particle interactions, in-situ TEM straining experiments were conducted on the Mo-La₂O₃ alloy, processed by rotary-swaging, from ambience to low temperature. It is observed that when an incoming dislocation interacts with the nanoparticle, multiple dislocation segments are generated. Those with one end pinned at the interface function as single-arm sources as shown in Fig. S11. While the dislocation segments with both ends pinned at the interface operate as Frank-Read dislocation sources. Snapshots in Fig. 4d (from Supplementary Movie 7) demonstrate a typical example showing the formation of Frank-Read dislocation sources and the subsequent generation of dislocations at room temperature. The Frank-Read dislocation segment pinned at the surface of particle 1, denoted as p1, bowed out and divided into two dislocation segments, denoted as d1′ and d1″, marked by yellow dashed curves. While d1′ detached from the particle later, the original Frank-Read dislocation source, denoted as d2 and indicated by blue dashed lines, continued to generate dislocations. At room temperature, the operation of these Frank-Read dislocation sources can persist through several cycles until the dislocation segments eventually detach from the particle-matrix interface. Such

interfacial Frank-Read dislocation sources are recreated as subsequent dislocations gliding towards and interacting with the particles.

The operation of these dislocation sources becomes more stable at low temperatures due to the increased difficulty of detachment. It is detected that during deformation at −50 °C, an abundance of short Frank-Read dislocation sources forms as incoming dislocations looped around the nanoparticles, as marked by the pink dashed squares in Fig. 4e. Figure 4f shows the operation of these Frank-Read dislocation sources (the series of snapshots is captured from Supplementary Movie 8). It is observed that when the dislocation, denoted as d3 and marked by the blue arrow, interacted with a particle (denoted as p2), one single-arm dislocation source and one Frank-Read dislocation source (F-R, pink arrows) formed. Subsequently, these sources commenced operation under applied stress, continuously generating dislocations. The operation of these dislocation sources proved to be highly durable. Even as the applied load increased significantly, these dislocation sources continued to operate stably, offering great benefits to the low-temperature dislocation plasticity of the material.

The aforementioned findings provide an approach to overcoming the strength-ductility trade-off in Mo alloy. The achievement originated from the diverse changes in the electronic density of states of rare-earth elements, which enables the formation of a "bumpy" and amorphous oxides-matrix interface under proper control of materials processing. This interface helps to divide one dislocation into many segments, creating dislocation sources pinned at the interface during dislocation-particle interactions. The dislocation sources created at the non-uniform interface exhibit the following characteristics: (a) they typically have a length on the order of tens of nanometers, a scale considered to be within the range of the most stable dislocation sources[39,40]; (b) the non-uniform interface captures dislocation

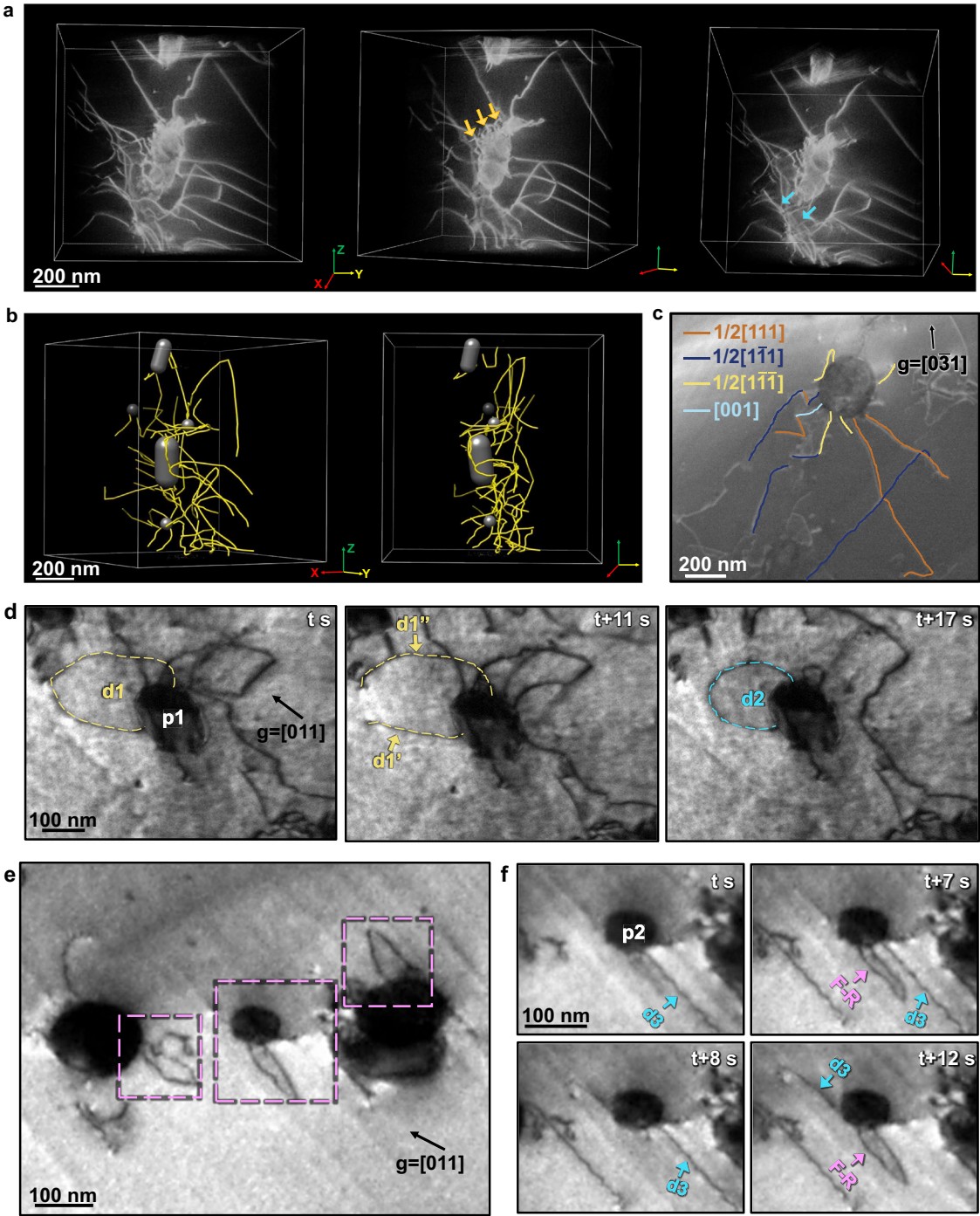

**Fig. 4 | The three-dimensional characterization and in-situ TEM observations on the dynamic behaviors of dislocation at different temperatures. a** The three-dimensional tomography of the particles and dislocations accumulated at the interface. The orange arrows indicated dislocation segments with both ends pinned at the interface. The blue arrows indicated the severe dislocation entanglements. **b** Images of three-dimensional dislocation structure reconstructed according to the experimental data. **c** g·b analysis. Two-beam dark-field TEM images with different diffraction vectors, **g**, annotating in each image with the direction shown by the black arrow. The dislocations with different Burgers vectors, **b** are colored accordingly. **d** The formation of Frank-Read dislocation sources at room temperature. Frank-Read dislocation source segment (d1, yellow dashed line) pinned at the surface of particle 1 (p1), divided into two segments (d1' and d1''), and subsequently, the original Frank-Read dislocation source (d2, blue dashed line) continued to generate dislocations. **e** The image of typical Frank-Read dislocation sources at −50 °C, as marked by pink dashed squares. **f** The series of TEM images showing the multiplication of dislocations via Frank-Read dislocation sources (F-R, pink arrows) when dislocation segment (d3, blue arrow) interacted with particle 2 (p2) at low temperature.

segments in random directions, leading to the further generation of various types of dislocations. The consequent dislocation interactions and multiplications culminate in the formation of a complex dislocation network, an accomplishment rarely seen in Mo alloys before. Such a mechanism empowers the material to achieve significant strength alongside a noticeable increase in ductility, even at low temperatures. The results not only advance our understanding of the fundamental mechanism of dislocation-particle interactions but also provide insights into the potential strategies for ameliorating the combination of high strength and high ductility of BCC materials.

## Methods

### Materials preparation

The sintered Mo-0.6 wt%$La_2O_3$ alloys were prepared by powered metallurgy, where the Mo-$La_2O_3$ alloy powders were derived from a liquid-solid mixing method[38]. The solution of lanthanum nitrate sprayed into solid molybdenum oxide powder was dried and given a heat treatment in hydrogen at 1050–1100 °C for 4.5 h to reduce molybdenum oxide and pyrolyze lanthanum nitrate. The powders were pressed to a cylindrical compact with a diameter of 90 mm via cold isostatic pressing and then sintered at 1850 °C for 4 h in the protection of dry hydrogen. The cylindrical compact was primarily forged into a rod with a diameter of 25 mm at 1250 °C, and rotary-swaged into a rod with a diameter of ~8 mm at successive decreasing temperatures from 1000 to 600 °C. Finally, the rods were annealed at 1100, 1200, 1400 °C, respectively, for 1 h.

### TEM sample preparation and microstructural characterization

**TEM sample preparation.** Samples for TEM characterization were first sectioned from the rod along the axial direction to 0.4 mm plates using electric discharge machining. Those plates were mechanically ground to 50 μm thickness and then punched into disks with a diameter of 3 mm. To achieve regions for observation, the samples were further thinned by twin-jet electropolishing using sulfuric acid solution, comprising 87.5% ethanol, 12.5% sulfuric acid at −15 °C and 15 V.

**Microstructural characterization.** The morphology observation of Mo-$La_2O_3$ alloy was carried out by a FEI Tecnai $G^2$ F20 S-TWIN operating at 200 kV. The high-resolution HADDF and iDPC images were obtained using a double Cs-corrected microscope (Thermo Fisher Spectra 300 (S)TEM), operated at 300 kV. The iDPC-STEM images were acquired using a 4-quadrant DF4 detector. The convergence angle was 25 mrad and the collection angle of iDPC-STEM imaging was set to 7-29 mrad. The EELS mappings were obtained by a JEOL JEM-ARM300F2 equipped with the Gatan K3 direct detection counting detector. The energy resolution as determined by measuring the full width at half-maximum of the ZLP was about 0.3 eV. EEL spectrum images were collected with a 12 mrad STEM semi-convergence angle and 130 mrad EELS collection angle. The map was collected using $62 \times 294$ probe positions with a 0.36 Å step size and with a dwell time of 10 ms.

**Three-dimensional tomography.** A double-tilt tomography holder (Fischione 2040) was used to acquire several tilts series of the HAADF-STEM images with an increment of 1°. For the tomography of different sizes of particles, the HAADF-STEM images were acquired at a tilting range of −60° to 76° (Fig. 2e) and -66° to 66° (Fig. S7), respectively. For the tomography of interaction between dislocations and particles, the HAADF-STEM images were acquired at a tilting range of −56° to 45° (Fig. 4a, b). In order to obtain the best contrast of dislocation, the diffraction vector (**g**) is set to [002], and the diffraction vector is guaranteed to remain constant during the tilt. After the image and axis alignment of the tomographic data in Tomviz software, the direct Fourier method was used to reconstruct the 3D tomography of particles and dislocations. The 3D reconstructed tomographic data was then visualized and rendered using the Volume Tracer tool in the Chimera software, as shown in Fig. 4b.

**In-situ TEM straining tests.** For in-situ TEM straining tests, the sample with a diameter of 3 mm was attached to 11.5 mm * 2.5 mm * 0.5 mm stainless-steel substrates. The stainless-steel substrates have a small hole in the middle for transmission of the electron beam. A Gatan 671 single-tilt straining holder was used to launch the in-situ tensile tests in the FEI Tecnai $G^2$ F20 microscope. The temperature of the holder is monitored by a calibrated silicon diode that provides a sensitive, linear temperature response. The conductor rod connecting the specimen holder tip to the liquid nitrogen dewar contains an electric heater to

change the specimen temperature. The sample should be stabilized for more than 30 minutes each time the temperature is changed. Tensile loading was accomplished by manually applying intermittent displacement pulses at a displacement rate of ~1 μm/s. Dislocation behavior was observed during the inter-pulse time.

### Molecular dynamics (MD) simulation

The MD simulation of BCC Mo containing amorphous MoPd particles was performed using the software package LAMMPS[41] with the empirical embedded-atom-method (EAM) potential[42]. The simulation cells are illustrated in Fig. 2b and 2c, with the dimensions of approximately 270 Å in $x$, 400 Å in $y$, and 400 Å in $z$, and contain 2.56 million atoms. Periodic boundary conditions (PBC) were imposed along $y$- and $z$-directions and shrink-wrapped non-periodic boundary conditions in the $x$-direction. The amorphous MoPd was firstly produced by melting at 5000 K within the simulation cell (250 Å for $x$, $y$, and $z$ direction with PBC), and then quenched to 300 K with the cooling rate of $10^{13}$ K/s. An amorphous MoPd particle with a diameter of 20 nm was cut and inserted into BCC Mo lattice (the overlapped atoms of Mo crystal were removed), see Fig. 2b. Two additional notches with a width of 50 Å and depth of 50 Å were applied on the amorphous sphere (see Fig. 2c). An edge dislocation line was then inserted in the simulation cell. After relaxation at 300 K for 1 ns, the loading setup follows that the bottom two layers of atoms in the $x$-direction were fixed and the top two layers of atoms were treated as a rigid body with the shear strain rate of $3.6 \times 10^{-2}$ /ns, while the NVT ensemble was applied to other atoms. The atomic configurations were displayed by OVITO[43].

## Data availability

All data generated or analyzed during this study are included in the published article and Supplementary Information and are available from the corresponding authors upon request.

## Code availability

All related codes are available from the corresponding authors on request.

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

## Acknowledgements

Q.Y. acknowledges support from the National Natural Science Foundation of China (Grant No. 52325102), Project supported by the Natural Science Foundation of Zhejiang province, China (Grant No. LZ22E010001), the National Key Research and Development Program of China under grant No. 2023YFB2405802, and the State Key Program for Basic Research in China under grant No. 2017YFA0208200. L.G. acknowledges support from the National Natural Science Foundation of China (52025025, 52250402). J.D. acknowledges support from the National Natural Science Foundation of China (Grant No. 12004294), the National Youth Talents Program and the HPC platform of Xi'an Jiaotong University. X.K. acknowledges the financial support of the National

Natural Science Foundation of China (12074017). Y.C. acknowledges financial support from the Shuimu Tsinghua Scholar Program. We acknowledge Dr. Jian Xu and Dr. Chuanhong Jin for their useful discussions.

## Author contributions

Q.Y., G.L., J.D. and L.G., proposed and supervised the research. Y.F. and Y.C. performed the microstructure analysis and corresponding data analysis including TEM, STEM, and in-situ TEM observation. P.C. synthesized the alloys and conducted the mechanical testing. J.D. and B.Z. carried out the molecular dynamics simulations. Y.C., Y.F., M.Z. and X.K. performed and analyzed the three-dimensional tomography of the particle and particle-dislocations interaction. Y.C., Y.F., J.Z. and Q.Z. analyzed the EELS data. Y.C. and Y.F. wrote the initial draft. All authors contributed to the discussion of the results and commented on the manuscript.

## Competing interests

The authors declare no competing interests.
