## [Peer Review File · Nature Communications]

The origin of exceptionally large ductility in Molybdenum Alloys dispersed with Irregular-Shaped La₂O₃ Nano-ParticlesREVIEWER COMMENTS

Reviewer #1 (Remarks to the Author):

This study reveals that incorporating lanthanum oxide particles with irregular shapes and amorphous interfaces into Mo through solid-liquid sintering and rotary-swaging leads to improved strength and ductility, particularly at low temperatures. The development of the advantageous amorphous interface is attributed to the distinctive valence states found in rare earth elements. The authors conducted high spatial, high-energy resolution, and high time resolution transmission electron microscopy (TEM) characterizations to elucidate this unique structure-property relationship. The findings of this research are intriguing and contribute to a better understanding of the significance of utilizing rare earth elements in alloying. With some minor revisions, the manuscript can be accepted by the journal.

1. The author compared the mechanical properties of commercially pure molybdenum with samples subjected to rotary swaging. It is recommended to also analyze the mechanical properties of samples in the sintered state.

2. Does the thickness of the amorphous interface change after rotary swaging? Or does the size of the particles change after rotary swaging?

3. BCC metals typically exhibit reduced plasticity at low temperatures due to the slower movement of screw dislocations and the fast movement of edge dislocations, making the interaction between screw and edge dislocations rare. Can the amorphous interface discussed in this paper positively influence the regulation of dislocation behavior in response to this mechanism?

4. What is the elemental distribution at the amorphous interface, and does it impact dislocation activities?

5. Do particles of different sizes interact with dislocations in the same way or differently?

6. The paper is well-written, but there are still some typos, for instance, ensure the correct notation of chemical formulas, such as La_2O_3 (line 65 in Supp); "dislocation tomography technique" in line 197...

7. The rotation axis is missing in the 3D dislocation tomography in Fig. 4a and Fig. 4b.

8. Have the authors tested the samples at even lower temperatures? Mo alloys usually display a ductile to brittle transition at room temperature. Therefore, it is of interest to know if the ductility can be maintained at even lower temperatures.

Reviewer #2 (Remarks to the Author):

In their investigation, the authors explored the interfacial structure between rare-earth oxide and the Mo matrix in a Mo-based alloy. They asserted that the interface structure plays a pivotal role in enhancing both the strength and ductility of the Mo alloy, given its capacity to modulate dislocation activities. The authors emphasized that the multivalent nature of rare-earth elements triggers the formation of an amorphous interface. Through the application of rotary swaging, they adeptly utilize this dual functionality, serving as pinning points and contributing to dislocation multiplication. This results in remarkable ductility and high strength of their Mo alloy at lower temperatures. The presented characterizations and discussions exhibit a high level of quality, making the results particularly relevant to the interface and BCC refractory alloys community. This manuscript can be accepted after incorporating the following revisions.

1. In line 119, where it mentions, "While other oxides, for instance, Al_2O_3 , usually display a

crystallized interface,". This is important for supporting that rare earth elements indeed facilitate the formation of an amorphous interface layer. Please provide references or other supporting evidence for this statement.

2. It would enhance reader understanding if the methods of rotary-swaging and 3D dislocation tomographic reconstruction are more explicitly described and referenced in the main text. For example, consider better linking these methods to the Method section or supporting material, as indicated in lines 159 and 202.

3. The presentation of data and methods is generally clear, with some potential improvements. For instance, in Fig. 1e, the oxygen vacancy and perfect crystal structure should be better labeled. Additionally, the color scale in Fig. 2a, particularly segment III, may be challenging to recognize and could benefit from clarification.

4. The coloration applied to the dislocation image obscures many details. Please consider making adjustments, particularly in Fig. 2a and Fig. 4d.

5. What is the size of the amorphous particle used in molecular dynamics simulation calculations? Please include a scale bar in the relevant figures. Additionally, explore whether the size of the particles influences dislocation-particle interactions.

6. The oxygen atoms were at times marked as pink balls and at other times as yellow balls. It is recommended to use a consistent color to prevent any potential confusion.

7. In Figure 1, why do some particles appear as black contrast in the HAADF image? Does the size of the particle change after rotary swaging? Consider addressing these points for clarification.

Response to Reviewers' Comments Letter

The origin of exceptionally large ductility in Molybdenum Alloys dispersed with Irregular-Shaped La_2O_3 Nano-Particles (manuscript NCOMMS-24-01038-T)

We sincerely thank the editor and all reviewers for their valuable review comments that we have used to improve the quality of our manuscript. The reviewer comments are laid out below and specific concern have been numbered, our point-to-point response is given in blue text and changes to the manuscript are given in red text.

Reviewer #1 (Remarks to the Author):

This study reveals that incorporating lanthanum oxide particles with irregular shapes and amorphous interfaces into Mo through solid-liquid sintering and rotary-swaging leads to improved strength and ductility, particularly at low temperatures. The development of the advantageous amorphous interface is attributed to the distinctive valence states found in rare earth elements. The authors conducted high spatial, high-energy resolution, and high time resolution transmission electron microscopy (TEM) characterizations to elucidate this unique structure-property relationship. The findings of this research are intriguing and contribute to a better understanding of the significance of utilizing rare earth elements in alloying. With some minor revisions, the manuscript can be accepted by the journal.

1. The author compared the mechanical properties of commercially pure molybdenum with samples subjected to rotary swaging. It is recommended to also analyze the mechanical properties of samples in the sintered state.

Response: We appreciate the referee's comment. Analyzing the mechanical properties of samples in the sintered state is indeed crucial. However, the pressureless sintered $\text{Mo-La}_2\text{O}_3$ alloy contains many casting flaws, resulting in relatively low density and medium performance. It may not accurately reflect the material's intrinsic mechanical properties. To address this limitation, we compared the mechanical properties of our $\text{Mo-La}_2\text{O}_3$ alloy treated with rotary-swaging to that treated by rolling (Mater. Sci. Eng. A 640, 320-329 (2015)), as illustrated in Fig. R1 and Fig. S9. Our findings confirm that the $\text{Mo-La}_2\text{O}_3$ alloy treated with rotary-swaging exhibits an excellent combination of strength and ductility.

Fig. R1 Mechanical properties of $\text{Mo-La}_2\text{O}_3$ alloy treated with rotary-swaging and rolling.

2. Does the thickness of the amorphous interface change after rotary swaging? Or does the size of the particles change after rotary swaging?

Response: As depicted in Fig. R2, the thickness of the amorphous layer remains essentially unchanged before and after rotary-swaging. Regarding the particle size, larger particles tend to fracture into smaller ones after rotary-swaging, facilitating the dispersion of nanoparticles. Consequently, the average particle size decreased. Furthermore, nanoparticles deformed into irregular shapes after rotary-swaging.

In the revised manuscript, we have included additional content in line 162: "Given the differential forces acting on specific crystallographic planes, large particles tend to fracture into smaller ones, and nanoparticles within the grain deform into irregular shapes."

Fig. R2 The HAADF-STEM images showing the amorphous interfacial layers of Mo-La₂O₃ alloys (a) in the sintered state and (b) after the treatment of rotary swaging.

3. BCC metals typically exhibit reduced plasticity at low temperatures due to the slower movement of screw dislocations and the fast movement of edge dislocations, making the interaction between screw and edge dislocations rare. Can the amorphous interface discussed in this paper positively influence the regulation of dislocation behavior in response to this mechanism?

Response: We appreciate the referee's comment. The amorphous layer discovered in our study indeed plays a crucial role in regulating dislocation behavior, particularly in enhancing the coordination between edge and screw dislocations. We observed that the presence of the amorphous layer effectively pins dislocations, resulting in the formation of multiple segments comprising different types of dislocations, including edge and screw dislocation components, at room temperature.

As shown in Fig. R3 (equivalent to the main text Fig. 2a), when an edge dislocation anchored by the amorphous layers continues to move, two screw dislocation segments emerge, significantly impeding the edge segments. This interaction promotes the cooperation of different dislocation components, facilitating various dislocation interactions and the operation of single-arm dislocation sources.

Moreover, this effect remains highly effective at lower temperatures. The pinning effect of the amorphous interface on dislocations becomes stronger, leading to the generation of stable Frank-Reed dislocation sources at low temperatures. This phenomenon greatly enhances the low-temperature dislocation plasticity of the material.

Fig. R3 The bright-field TEM images showing the cooperation between screw and edge dislocation segments induced by the amorphous interface.

4. What is the elemental distribution at the amorphous interface, and does it impact dislocation activities?

Response: The EDS mapping of the amorphous interface is presented in Fig. R4. Our analysis revealed that the elements present in the amorphous layer are primarily La and O, providing critical evidence that the amorphous layer originates from the structural evolution of La_2O_3 . Interestingly, we observed that the relative concentration of oxygen at the amorphous interface is slightly lower than that in the particle's interior, indicating the presence of oxygen vacancies.

Furthermore, the distribution of different elements in the amorphous interface layer appears to be relatively uniform. Therefore, we believe that it is the amorphous structure of the interfacial layer that predominantly influences the dislocation activities and material properties, rather than the specific elemental distribution.

Fig. R4 The EDS mapping showing the elemental distribution at the amorphous interface.

5. Do particles of different sizes interact with dislocations in the same way or differently?

Response: We appreciate the referee's feedback. Our study indeed found that particles of different sizes interact with dislocations in a similar manner. We focused on the interaction between dispersed La_2O_3 particles and dislocations within the grain, with nanoparticle sizes ranging from 20 nm to 500 nm. Due to the presence of the amorphous layer, both large and small particles exhibited a strong pinning effect on dislocations. This effect facilitated interactions between dislocations and the formation of various types of dislocation sources. However, particles of different sizes showed variations in their ability to accumulate dislocations. As illustrated in Fig. R5, both particle 1 and

particle 2 exhibited a significant pinning effect on dislocations, leading to the formation of single-arm or Frank-Read dislocation sources on the surface of the particles. Importantly, larger particles resulted in greater dislocation accumulation.

Fig. R5 The HAADF-STEM image showing dislocation-particle interaction with different particle sizes. (a) smaller particle (<100 nm); (b) larger particle (>200 nm)

6. The paper is well-written, but there are still some typos, for instance, ensure the correct notation of chemical formulas, such as La_2O_3 (line 65 in Supp); “dislocation tomography technique” in line 197...

Response: We sincerely apologize for the inconvenience of reading caused by the typos. Accordingly, we have changed “ La_2O_3 ” to “ La_2O_3 ” (line 65 in Supp), and “dislocation tomography techniques” to “dislocation tomography technique” (line 197). We have also corrected other typos throughout the manuscript.

7. The rotation axis is missing in the 3D dislocation tomography in Fig. 4a and Fig. 4b.

Response: We appreciate the reviewer for pointing out it. In Fig. 4a and b in the revised manuscript, we have added annotations indicating the rotation axis at the corresponding positions, as shown here in Fig. R6.

Fig. R6 The three-dimensional characterization. (a) The three-dimensional tomography of the particles and dislocations accumulated at the interface. (b) Image of three-dimensional dislocation structure stimulated according to the experimental data.

8. Have the authors tested the samples at even lower temperatures? Mo alloys usually display a ductile to brittle transition at room temperature. Therefore, it is of interest to know if the ductility

can be maintained at even lower temperatures.

Response: We appreciate the referee's interest in exploring the material's ductility at even lower temperatures. To address this, we conducted tensile tests on Mo-La₂O₃ samples after rotary swaging at -50 and -90 °C. The tensile samples were dog-bone shaped with gauge sizes of 25 mm in length and 5 mm in diameter. Fig. R7 illustrates the tensile engineering stress-strain curves of the tested alloys at -50 and -90 °C. The samples experienced fracture failure at approximately 10% ductility at -90 °C, suggesting that the material still maintains decent ductility at such low temperatures.

Fig. R7 The engineering strain-stress curves of Mo-La₂O₃ alloy at -50 (red line) and -90 °C (blue line)

Reviewer #2 (Remarks to the Author):

In their investigation, the authors explored the interfacial structure between rare-earth oxide and the Mo matrix in a Mo-based alloy. They asserted that the interface structure plays a pivotal role in enhancing both the strength and ductility of the Mo alloy, given its capacity to modulate dislocation activities. The authors emphasized that the multivalent nature of rare-earth elements triggers the formation of an amorphous interface. Through the application of rotary swaging, they adeptly utilize this dual functionality, serving as pinning points and contributing to dislocation multiplication. This results in remarkable ductility and high strength of their Mo alloy at lower temperatures. The presented characterizations and discussions exhibit a high level of quality, making the results particularly relevant to the interface and BCC refractory alloys community. This manuscript can be accepted after incorporating the following revisions.

1. In line 119, where it mentions, "While other oxides, for instance, Al₂O₃, usually display a crystallized interface,". This is important for supporting that rare earth elements indeed facilitate the formation of an amorphous interface layer. Please provide references or other supporting evidence for this statement.

Response: We appreciate the referee's suggestion and have incorporated the reference into the manuscript to reinforce the argument that Al₂O₃ typically displays a crystallized interface (J. Alloys Compd. 823, 153748 (2020)). In our study, we characterized the interfacial structure between Al₂O₃ particles and the Mo matrix. As shown in Fig. R8, a typical incoherent interface was observed. However, both the STEM image and the FFT pattern provided no evidence for the existence of an amorphous layer. We have also included this information as Fig. S4 in the supplementary

information to further clarify this point.

Fig. R8 The STEM image and the corresponding FFT pattern showing the interfacial structure between Mo matrix and Al_2O_3 particle.

2. It would enhance reader understanding if the methods of rotary-swaging and 3D dislocation tomographic reconstruction are more explicitly described and referenced in the main text. For example, consider better linking these methods to the Method section or supporting material, as indicated in lines 159 and 202.

Response: We appreciate the referee's advice, and we have incorporated the corresponding content into the revised manuscript. For instance, in line 159, we added the description "... The sintered $\text{Mo-La}_2\text{O}_3$ was rotary-swaged into a rod at successive decreasing temperatures from 1000 to 600 °C and annealed at 1350 °C, which subjects the material to compression in all directions, inducing substantial deformation in both the particle and the Mo matrix (detailed in Methods). "

In lines 202 and 211, we added the description "... The series of HAADF-STEM images was captured by using a double-tilt tomography holder at a tilting range for -56° to 45° with an increment of 1° .", "...By plugging in the 3D tomography of the particle and dislocations, we attained a simulated 3D dislocation structure, as displayed in Fig. 4b."

3. The presentation of data and methods is generally clear, with some potential improvements. For instance, in Fig. 1e, the oxygen vacancy and perfect crystal structure should be better labeled. Additionally, the color scale in Fig. 2a, particularly segment III, may be challenging to recognize and could benefit from clarification.

Response: We thank the referee for the useful suggestions. We have modified the Fig. 1e and 2a in the revised manuscript, as displayed in Fig. R9 and Fig. R10, respectively.

Fig. R9 The iDPC-STEM image of the atomic structure in the interface displaying (a) the oxygen vacancy in La_2O_3 and (b) the oxygen clustering in Mo matrix near the La_2O_3 particle.

Fig. R10 The TEM images captured from the movie showing the cooperation between screw and edge dislocation segments induced by the amorphous layers in Mo-La₂O₃ interface.

4. The coloration applied to the dislocation image obscures many details. Please consider making adjustments, particularly in Fig. 2a and Fig. 4d.

Response: We thank the referee for the useful suggestions. We have modified the Fig. 2a and 4d in the revised manuscript, as displayed in Fig. R10 and Fig. R11, respectively.

Fig. R11 The series of TEM images captured from the movie showing the formation of Frank-Read dislocation sources at room temperature.

5. What is the size of the amorphous particle used in molecular dynamics simulation calculations? Please include a scale bar in the relevant figures. Additionally, explore whether the size of the particles influences dislocation-particle interactions.

Response: We thank the referee for the useful suggestions. The diameter of the amorphous particle used in molecular dynamics simulation calculations is 20 nm. We have added scale bar in Fig. 2b and c in the manuscript, as displayed in Fig. R12.

Fig. R12 Molecular dynamics simulation showing the interaction between amorphous particle and dislocations.

We observed that particles of different sizes interacted with dislocations in a similar manner. Our study focused on the interaction between dispersed La₂O₃ particles and dislocations within the

grain, with nanoparticle sizes ranging from 20 nm to 500 nm.

Due to the presence of the amorphous layer, both large and small particles exhibited a strong pinning effect on dislocations. This effect facilitated interactions between dislocations and the formation of various types of dislocation sources. However, particles of different sizes showed variations in their ability to accumulate dislocations.

As illustrated in Fig. R4 (in the response to Question 2 from reviewer 1), both particle 1 and particle 2 exhibited a significant pinning effect on dislocations, leading to the formation of single-arm or Frank-Read dislocation sources on the surface of the particles. Importantly, larger particles resulted in greater dislocation accumulation.

6. The oxygen atoms were at times marked as pink balls and at other times as yellow balls. It is recommended to use a consistent color to prevent any potential confusion.

Response: We thank the referee for the suggestion. In the revised manuscript, we used yellow color to indicate the oxygen atoms in Fig. 1e, as displayed in Fig. R9.

7. In Figure 1, why do some particles appear as black contrast in the HAADF image? Does the size of the particle change after rotary swaging? Consider addressing these points for clarification.

Response: The black contrast in the HAADF-STEM image actually indicates the presence of holes where particles were missing during the sample preparation process for TEM and STEM observation.

As described in the response to Question 2 from reviewer 1, the size of large particles, especially those dispersed within the grain, changes following rotary swaging. Large particles tend to fracture into smaller ones after rotary forging, which aids in the dispersion of nanoparticles, resulting in a decrease in the average particle size.

We appreciate the referee's valuable advice. In line 162 of the manuscript, we have added further clarification: "...Given the relative forces acting on specific crystallographic planes may differ, large particles tend to fracture into smaller ones and the nanoparticles within the grain are deformed into irregular shapes."

REVIEWERS' COMMENTS

Reviewer #1 (Remarks to the Author):

All my questions have been responded properly, the manuscript can be accepted.

Reviewer #2 (Remarks to the Author):

The authors address my comments well, and this manuscript can be accepted now.